# The Combination of Patient-Reported Clinical Symptoms and an Endoscopic Score Correlates Well with Health-Related Quality of Life in Patients with Ulcerative Colitis

**DOI:** 10.3390/jcm8081171

**Published:** 2019-08-05

**Authors:** Satimai Aniwan, David H. Bruining, Sang Hyoung Park, Badr Al-Bawardy, Sunanda V. Kane, Nayantara Coelho Prabhu, John B. Kisiel, Laura E. Raffals, Konstantinos A. Papadakis, Darrell S. Pardi, William J. Tremaine, Edward V. Loftus

**Affiliations:** 1Division of Gastroenterology and Hepatology, Mayo Clinic, Rochester, MN 55905, USA; 2Division of Gastroenterology, Chulalongkorn University, King Chulalongkorn Memorial Hospital, Thai Red Cross Society, Bangkok 10330, Thailand; 3Department of Gastroenterology, University of Ulsan College of Medicine, Asan Medical Center, Songpa-gu, Seoul 138-736, Korea

**Keywords:** ulcerative colitis, outcome, quality of life, endoscopy

## Abstract

Background and aims: Patient-reported outcomes (PROs) will become increasingly important as primary endpoints in future clinical trials. We aimed to evaluate the relationship between health-related quality of life (HRQoL) and the combination of patient-reported clinical symptoms (ClinPRO2) and Mayo endoscopic subscore (MES) in patients with ulcerative colitis (UC). Methods: We conducted a prospective cross-sectional study of 90 consecutive UC patients who were scheduled for sigmoidoscopy or colonoscopy. All patients completed the following questionnaires: (1) self-rated rectal bleeding and stool frequency (ClinPRO2); (2) Short Inflammatory Bowel Disease Questionnaire (SIBDQ); (3) European Quality of Life 5-Dimensions 3-Level (EQ5D3L); (4) Work Productivity and Activity Impairment questionnaire (WPAI); (5) Functional Assessment of Chronic Illness Therapy-Fatigue (FACIT-F); and (6) Hospital Anxiety and Depression Scale (HADS). The endoscopic images were graded according to the MES. “No symptoms” was defined as a symptom score of 0, and “mucosal healing” was defined as MES score of 0–1. Correlations between the combined ClinPRO2 and MES with HRQoL were assessed using Spearman’s correlation coefficients. Results: The combination of the ClinPRO2 and MES was well correlated to SIBDQ (*r* = −0.70), EQ5D3L (*r* = −0.51), WPAI (*r* = 0.62), FACIT-F (*r* = −0.58), and HADS-depression (*r* = 0.45). SIBDQ scores had strong correlations with FACIT-F (*r* = 0.86), WPAI (*r* = −0.80), and HADS-depression (*r* = −0.75) (*p* < 0.05 for all correlations). Patients with no symptoms reported the greatest all HRQoL scores. Conclusions: In patients with ulcerative colitis, the combination of a ClinPRO2 and the MES had good correlation with the SIBDQ. In addition, SIBDQ was well correlated to the various HRQoL.

## 1. Introduction

Ulcerative colitis (UC) is characterized by chronic inflammation involving the colon mucosa. The natural course of UC is relapsing and remitting [1]. Uncontrolled bowel inflammation leads to an increased risk of gastrointestinal complications [2]. Currently there are no curative medical treatments for UC. Over the past decade, the goal of UC management has focused on disease outcomes such as disease activity, disease-related hospitalization, and disease-related surgery through the use of novel inflammatory bowel disease (IBD) drugs, including biologic agents.

Previously, the primary endpoints in clinical trials were composite disease activity indexes, comprised of symptoms reported by patients, well-being rated by the physician, and laboratory and/or endoscopic findings. These composite indexes, however, may no longer be accepted for drug approval by the United States Food and Drug Administration (FDA) [3], because individual items of the disease activity indexes are difficult to weigh equally and physician rating of well-being is very subjective. In addition, some UC symptoms, including stool frequency and stool urgency, are similar to symptoms of irritable bowel syndrome (IBS), which may be present in up to one-third of inactive UC patients [4]. Consequently, the criteria for inclusion in a treatment trial intended for FDA review for new drug approval for treatment of UC have changed—participants must have both colonic symptoms plus active colonic mucosal inflammation [5]. The FDA has proposed multiple endpoints, including a pure patient-reported outcomes (PROs) endpoint and objective measurements of disease. In addition, FDA guidance for developing PROs indicate that the outcomes reported by patients should focus on health-related quality of life (HRQoL) including work productivity, fatigue, and mood [3].

Therapeutic targets for “treat-to-target” UC management were recently recommended by the International Organization for the Study of Inflammatory Bowel Disease (IOIBD) [6]. The co-primary endpoints were defined as the combination of symptoms, with the desired target being resolution of rectal bleeding and normalization of bowel habit (ClinPRO2), and endoscopic remission, with the desired target being a Mayo endoscopic subscore (MES) of 0–1 [6]. Various PRO instruments measuring HRQoL have been used as secondary outcomes in many clinical trials of IBD; however, data on the relationships between symptoms and endoscopic targets and these PROs are limited.

The aims of this study were: (1) to evaluate the relationship between the combination of ClinPRO2 and MES with HRQoL measuring quality of life, work productivity, fatigue, and mood; and (2) to evaluate the relationship between the Short Inflammatory Bowel Disease Questionnaire (SIBDQ) and these HRQoL.

## 2. Materials and Methods

### 2.1. Study Population

We conducted a prospective cross-sectional study including consecutive UC patients who attended the IBD Clinic in the Division of Gastroenterology and Hepatology at Mayo Clinic, Rochester, Minnesota and were scheduled for clinically indicated sigmoidoscopy or colonoscopy. UC was diagnosed according to previous well-defined criteria [7,8,9], based on the following findings on 2 occasions separated by at least 6 months; diffused granular or friable colonic mucosa on endoscopy; and continuous mucosal involvement based on endoscopy or barium studies [7,8,9]. Patients with primary sclerosing cholangitis, spondyloarthropathy, active cancer, previous pelvic radiation, partial colectomy, or proctocolectomy and ileal pouch-anal anastomosis were excluded. All patients provided written informed consent. This study was approved by the Institutional Review Board of Mayo Clinic.

### 2.2. Symptoms and Patient-Reported Outcomes

Within 7 days before or after the endoscopy, all patients completed the following validated questionnaires: (1) self-rated rectal bleeding and stool frequency using the 6-point partial Mayo score (ClinPRO2) [10]; (2) SIBDQ [11,12]; (3) European Quality of Life 5-Dimensions 3-Level, time-trade-off (EQ5D3L, TTO) [13]; (4) Work Productivity and Activity Impairment questionnaire (WPAI) [14]; (5) Functional Assessment of Chronic Illness Therapy-Fatigue (FACIT-F) [15]; and (6) Hospital Anxiety and Depression Scale (HADS), as shown in Appendix A [16].

The SIBDQ is a 10-item disease-specific HRQoL instrument measuring four domains—bowel, emotion, social activities, and general well-being, as shown in Appendix A. Each item has a 7-point scale from 1 to 7. The absolute SIBDQ score ranges from 10 to 70, with higher values indicating better QoL. The EQ5D3L questionnaire is a simple generic health utility instrument measuring 5 dimensions of health—mobility, self-care, usual activities, pain/discomfort, and anxiety/depression, as shown in Appendix A. Each dimension has a 3-level scale from 1 (no problem) to 3 (extreme problem). A 5-digit number of these 5-dimension 3-level scores was then converted to a United States TTO score, for example, “11111” corresponding to a TTO score of 1.000 indicating the best QoL and “33333” corresponding to a TTO score of −0.109 indicating the worst QoL. The EQ5D3L-TTO score for the US ranges from −0.109 to 1.000, with higher scales indicating better QoL. The WPAI questionnaire is a 6-question work assessment regarding employment, hours missed due to health problems, hours actually working, and degree of health-affected productivity while working and while doing regular activity using a 0–10 visual analogue scale, as shown in Appendix A. Percentage overall work impairment ranges from 0% to 100%, with higher scales indicating poor work productivity. The FACIT-F is a 13-question measurement to assess fatigue associated with chronic illness, as shown in Appendix A. Each question has a 5-point scale from 0 to 4. FACIT-F scores range from 0 to 52, with higher values indicating less fatigue. The HADS consists of a 7-question depression assessment and 7-question anxiety assessment, as shown in Appendix A. Each subscore for depression and anxiety ranges from 0 to 21, with higher scores indicating a higher degree of depression/anxiety.

### 2.3. Endoscopic Inflammation

Two expert endoscopists (E.V.L. and D.H.B.), who were blinded to the results of these questionnaires, graded the endoscopic still images independently using the Mayo endoscopic subscore (MES) ranging from 0 to 3: (0) normal; (1) erythema, decreased vascular pattern; (2) marked erythema, absent vascular pattern, friability, erosion; and (3) spontaneous bleeding, ulceration. The endoscopic inflammation was graded in the most severely affected segment of bowel. Both endoscopists discussed discordant cases to reach a consensus.

### 2.4. Disease Activity According to the Symptoms and MES

“No symptoms” was defined as a 6-point Mayo symptom score of 0 (i.e., no rectal bleeding and normal stool frequency). Mucosal healing (MH) was defined as MES score of 0-1. UC patients were categorized into 3 groups as: group (1) no symptoms; group (2) symptoms present but MH; and group (3) symptoms present and no MH.

### 2.5. Statistical Analysis

With the assumption that the correlation coefficients (*r*-values) between the combination of ClinPRO2 and MES with HRQoL and those between SIBDQ and various HRQoL were greater than or equal to 0.3, a sample size of at least 90 UC patients was required to detect these differences with a power of 80% at a two-sided significance level of 0.05. Continuous data were reported as means with standard deviations (SD) or medians with interquartile ranges (IQR). Categorical data were reported as counts and proportions. The correlations were assessed using Spearman’s correlation coefficients. We defined the *r*-value of ≥0.7 as a strong correlation, 0.50–0.69 for a moderate correlation, 0.30–0.49 for a fair correlation, and <0.3 for a poor correlation [17]. To compare HRQoL according to ClinPRO2 and mucosal healing, the analysis of variance (ANOVA) or the Kruskal–Wallis test was used and post hoc analysis for multiple comparisons with the Bonferroni method was used. An alpha-level of 0.05 was considered as statistically significant. Statistical analyses were performed by using JMP 12 statistical software package (SAS Institute Inc., Cary, NC, USA).

## 3. Results

A total of 90 UC patients were included in this study. The median age was 45 years (IQR, 32–56) and 46% were male. Disease extent was extensive colitis in 55 patients (61%), left-sided colitis in 25 (28%), and proctitis/proctosigmoiditis in 10 patients (11%). No symptoms were reported by 25 patients (28%). Mucosal healing was seen in 36 patients (40%). Overall, median SIBDQ, EQ5D3L-TTO, WPAI, FACIT-F, HADS-depression, and HADS-anxiety scores were 48.5, 0.827, 36%, 33.1, 4, and 5, respectively. Demographic data including patient characteristics, ClinPRO2, MES, and HRQoL are shown in Table 1.

### 3.1. Correlations of Symptoms, MES, and PROs

Table 2 shows all correlations among ClinPRO2, MES, and HRQoL. The MES alone was fairly correlated to SIBDQ (*r* = −0.47), WPAI (*r* = 0.44), FACIT-F (*r* = −0.34), and EQ5D3L (*r* = −0.31). The MES alone was poorly correlated to HADS-depression (r = 0.22) and HADS-anxiety (*r* = 0.22). ClinPRO2 alone correlated strongly with SIBDQ (*r* = −0.73), and moderately with WPAI (*r* = 0.63), FACIT-F (*r* = −0.62), EQ5D3L (*r* = −0.53), and HADS-depression (*r* = 0.50). The combination of symptoms and MES was strongly correlated to SIBDQ (r = −0.70), as shown in Figure 1A, and modestly correlated to EQ5D3L (*r* = −0.51), WPAI (*r* = 0.62), and FACIT-F (*r* = −0.58). There were fair correlations between the combination of symptoms and MES with HADS-depression (*r* = 0.45) and HADS-anxiety (*r* = 0.30) (*p* < 0.05 for all correlations).

### 3.2. Correlations of SIBDQ and Various PROs

SIBDQ scores had strong correlations with fatigue (FACIT-F) (*r* = 0.86), as shown in Figure 1B, work impairment (WPAI) (*r* = −0.80), and HADS-depression (*r* = −0.75). SIBDQ scores had modest correlations with EQ5D3L (*r* = 0.67) and HADS-anxiety (*r* = −0.53), *p* < 0.01 for all correlations, as shown in Table 2.

ClinPRO2: the self-rated rectal bleeding and stool frequency using 6-points of the partial Mayo score, MES: Mayo endoscopic subscore, SIBDQ: Short Inflammatory Bowel Disease Questionnaire, EQ5D3L: European Quality of Life 5-Dimensions 3-Level (time-trade-off), WPAI: Work Productivity and Activity Impairment questionnaire (% overall work productivity impairment), FACIT-F: Functional Assessment of Chronic Illness Therapy-Fatigue, HADS: Hospital Anxiety and Depression Scale.

### 3.3. Association between Disease Activity and PROs

Twenty-five UC patients (28%) had no symptoms 15 (17%) had symptoms present but MH, and 50 patients (55%) had symptoms present and no MH. There were no significant differences in age, sex, UC extent, or medication use among the three groups. There were significant differences in the scores of SIBDQ, EQ5D3L, WPAI, FACIT-F, HADS-depression, and HADS-anxiety among the three groups (*p* < 0.01 for all comparisons), as shown in Table 3. Comparisons between two groups, both symptomatic UC patients with or without MH, had significantly experienced poorer all HRQoL than these with no symptoms, whereas all HRQoL scores were similar between symptomatic patients but MH and symptomatic patients with no MH, as shown in Figure 2.

## 4. Discussion

The results of this study of consecutive UC patients assessed in an IBD clinic demonstrate that the combination of ClinPRO2 and MES correlated well with both disease-specific and generic measures of HRQoL (i.e., SIBDQ, EQ5D3L), work productivity, and fatigue, whereas there were fair relationships between the combination of ClinPRO2 and MES and depression and anxiety as measured by HADS. In addition, SIBDQ was strongly correlated to the various PROs with regard to work productivity, fatigue, depression, and anxiety.

Several studies have demonstrated relationships between disease activity indexes and HRQoL [15,16,18]. Theede et al. evaluated the relationship between the Simple Clinical Colitis Activity Index (SCCAI), MES, and SIBDQ among 110 UC patients. The authors showed that SCCAI and MES alone were significantly correlated to SIBDQ scores (*r* = −0.79 and *r* = −0.58, respectively) [18]. Nevertheless, in a subgroup analysis of UC patients with clinical remission (SCCAI ≤ 1), there was no difference in SIBDQ scores between UC patients with and without MH using a definition of MES score of 0 (63 vs. 62) [18]. Recently, Gracie and coworkers described a modest association between SCCAI and depression but not anxiety [19]. Higher HADS-depression scales were associated with UC patients with clinically active disease (SCCAI ≥ 5) (odds ratio, 1.21 per 1-point HADS increase) [19]. A study from Germany demonstrated a modest relationship between disease activity as measured by the CAI and the EQ5D3L-TTO in UC patients (*r* = −0.67, *p* < 0.01) [13]. These results seem consistent that achieving clinical remission is associated with good HRQoL. However, when moving the therapeutic target from a composite disease activity index to co-primary endpoints of clinical and endoscopic targets, the relationship with HRQoL needs to be re-evaluated.

In our study, we used a pure patient-reported symptom index instead of a traditional composite disease activity index and applied the standard definition of MH from the proposed endoscopic target [6]. Our results are in line with those previous studies and go even further, showing the importance of achieving “no symptoms” in order to obtain better HRQoL, not only as measured by the disease-specific SIBDQ, but also the more generic EQ5D3L, and other important PROs assessing work productivity, fatigue, and mood. Using a combination of ClinPRO2 and MES, a difference of 21.5 points in SIBDQ scores was observed when comparing UC patients with symptoms present and no MH to those with no symptoms. Similarly, median FACIT-F scales were 18.5 points higher among patients with symptoms present and no MH than those with no symptoms. With respect to generic HRQoL, patients with symptoms present and no MH had lower median EQ5D3L scores than that of the US general population [20] (0.816 vs. 0.867, *p* < 0.01). In contrast, patients with no symptoms reported higher median EQ5D3L scores than the US general population (1 vs. 0.867, *p* < 0.01).

Interestingly, we found poor correlations between MES alone with HRQoL but good relationships between ClinPRO2 alone with HRQoL. The correlation (*r*-value) when combining ClinPRO2 and MES with HRQoL appears numerically a little lower than those observed with ClinPRO2 only. We could hypothesize that beyond FDA recommendations, ClinPRO2 mostly correlated with functional bowel symptoms, raising some questions about the use of symptom scores only. Our study highlights the importance of using combined multiple endpoints including colonic symptoms, PROs, and objective evidence of inflammation. In a subgroup of UC patients with symptoms, there were no significant differences in all HRQoL regardless of mucosal healing.

The SIBDQ, a short form of the IBD questionnaire (IBDQ), is a simple 10-item questionnaire which takes approximately 5 min to complete. The SIBDQ identifies 90% of the IBDQ in UC patients [11]. It has been developed for community physicians to assess the effect of IBD on the patient’s life. Previous studies have demonstrated the validity, reliability, and responsiveness of SIBDQ in UC patients [11,21]. A mean decrease of 11.8 points correlates with important clinical changes defined by the physician; from remission to mild relapse and mild to moderate relapse [21]. The relationship between SIBDQ and other aspects of health status has only been explored briefly. The Crohn’s and Colitis Foundation of America Partners’ Internet-based cohort study showed the association between SIBDQ and the PRO Measurement Information System (PROMIS; fatigue, social role satisfaction, sleep disturbance, depression, anxiety, and pain) [12]. Although PROMIS is a general measure for patients with chronic illness, it addresses broad symptoms and functions. In contrast, our study used the specific instruments for each health domain of interests addressing a comprehensive set of disease-specific symptoms. Our results are consistent with the Internet-based cohort [12]. SIBDQ scores had very high correlations with FACIT-F, WPAI, and HADS-depression scores, suggesting that SIBDQ can capture many important aspects of health status.

A key strength of this study is that all patients provided the self-administered symptoms, SIBDQ, various HRQoL questionnaires, and underwent endoscopy during the same period of time. Therefore, we were able to verify the disease activity according to symptoms and objective evidence of inflammation and compare them across various HRQoL. Nevertheless, we acknowledge some limitations. First, this was a cross-sectional study. Our data supports only how the co-primary endpoints influence the HRQoL in UC patients. Further validated studies from other centers and other geographic regions would be useful. Additionally, further longitudinal studies are needed to explore whether improving disease activity improves the quality of life. Second, we initially attempted to assess the HRQoL of patients with no symptoms (MH and no MH). Unfortunately, only four patients with no symptoms did not have MH. This may limit the strength of the conclusion. Third, we did not evaluate other markers of inflammation such as C-reactive protein and fecal calprotectin. Fourth, the HRQoL questionnaires used in this study did not meet the FDA guidance for PRO development. Ideally, PROs could be any direct reports from individual patients about their health condition and its treatment without any interpretation by a physician [3]. Unfortunately, none of the current available HRQoL questionnaires in IBD meet that FDA guidance [3]. Fifth, the majority of our study population was Caucasian. The generalizability of these data to different racial and ethnic populations may not be possible. Lastly, the lack of central reading for endoscopic findings could have introduced some bias; however, our two independent endoscopic readers were blinded to the clinical symptoms and HRQoL scores.

In conclusion, the combination of ClinPRO2 and MES had good correlation with SIBDQ in UC. SIBDQ was well correlated to the various HRQoL. Furthermore, SIBDQ and various HRQoL questionnaires can differentiate patients with or without active disease according to clinical and endoscopic targets. The SIBDQ is therefore a useful tool which could be incorporated into clinical research and practice. However, before the implementation of ClinPRO2 and HRQoL as a primary endpoint in clinical trials, the development of a reliable measurement of PROs from the patient’s perspective is necessary and requires future longitudinal studies from multinational regions.

## Figures and Tables

**Figure 1 jcm-08-01171-f001:**
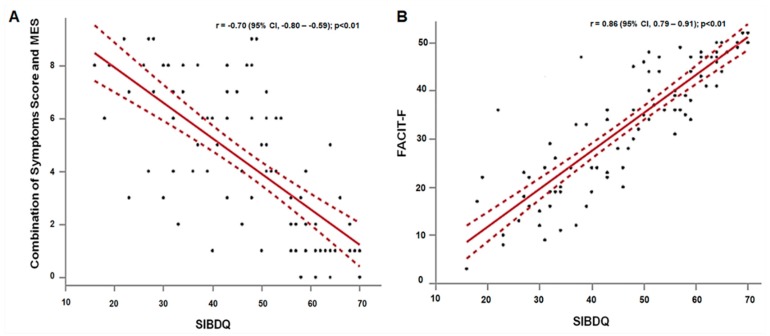
Correlations of (**A**) combination of ClinPRO2 and MES with SIBDQ and (**B**) SIBDQ with FACIT-F.

**Figure 2 jcm-08-01171-f002:**
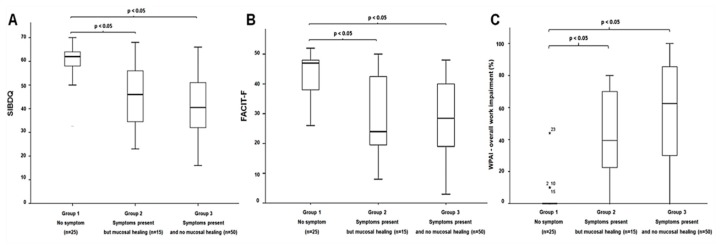
(A) SIBDQ, (B) FACIT-F, and (C) WPAI in UC patients categorized by ClinPRO2 and mucosal healing.

**Table 1 jcm-08-01171-t001:** Demographic and clinical characteristics of 90 patients with ulcerative colitis.

Variables	N = 90
Median age, years (IQR)	45 (32–56)
Males, n (%)	41 (46%)
White race, n (%)	86 (96%)
Non-Hispanic ethnicity, n (%)	90 (100%)
Mean body mass index, kg/m^2^ (SD)	27.9 (6.8)
Smoking status, n (%)	
Never-smoker	60 (67%)
Ex-smoker	28 (31%)
Current smoker	2 (2%)
Other comorbid diseases, n (%)	
Diabetes mellitus	4 (4%)
Hypertension	5 (6%)
None	81 (90%)
Median UC duration, years (IQR)	9.9 (4.2–17.2)
UC location, n (%)	
Proctitis/proctosigmoiditis	10 (11%)
Left-sided colitis	25 (28%)
Extensive colitis	55 (61%)
UC current medication, n (%)	
5-Aminosalicylates	38 (42%)
Immunomodulators	25 (28%)
Systemic corticosteroids	16 (18%)
Biologic agents	40 (44%)
None	10 (11%)
ClinPRO2	
Rectal bleeding, n (%)	
Score 0: no blood seen	46 (51%)
Score 1: streaks of blood less than half the time	20 (22%)
Score 2: obvious blood most of the time	17 (19%)
Score 3: blood passes alone	7 (8%)
Stool frequency, n (%)	
Score 0: normal	27 (30%)
Score 1: 1–2 per day more than normal	17 (19%)
Score 2: 3–4 per day more than normal	14 (16%)
Score 3: ≥5 per day more than normal	32 (35%)
Mayo endoscopic subscore, n (%)	
Score 0: normal	10 (11%)
Score 1: erythema, decreased vascular pattern	26 (29%)
Score 2: marked erythema, absent vascular pattern, friability, erosion	18 (20%)
Score 3: spontaneous bleedings, ulceration	36 (40%)
Median SIBDQ (IQR)	48.5 (34–59)
Median EQ5D3L-TTO (IQR)	0.827 (0.778–1)
Median WPAI (IQR)	
% overall work impairment	36 (0–80)
Median FACIT-F (IQR)	33.1 (13.1)
Median HADS-Depression (IQR)	4 (1–7)
Median HADS-Anxiety (IQR)	5 (3–9)

UC, ulcerative colitis; IQR, interquartile range; SD, standard deviation; ClinPRO2, the self- rated rectal bleeding and stool frequency using 6-points of the partial Mayo score; SIBDQ, Short Inflammatory Bowel Disease Questionnaire; EQ5D3L-TTO, European Quality of Life 5-Dimensions 3-Level, time-trade-off; WPAI, Work Productivity and Activity Impairment questionnaire; FACIT-F, Functional Assessment of Chronic Illness Therapy-Fatigue; HADS, Hospital Anxiety and Depression Scale.

**Table 2 jcm-08-01171-t002:** Spearman’s correlation between ClinPRO2, Mayo endoscopic subscore, and patient-reported outcomes (PROs).

Outcomes	SIBDQ	EQ5D3L	WPAI	FACIT-F	HADS Depression	HADS Anxiety
**MES**	−0.47 *	−0.31 *	0.44 *	−0.34 *	0.22 ^†^	0.22 ^†^
**ClinPRO2**	−0.73 *	−0.53 *	0.63 *	−0.62 *	0.50 *	0.30 *
**Combination of ClinPRO2 and MES**	−0.70 *	−0.51 *	0.62 *	−0.58 *	0.45*	0.30 *
**SIBDQ**	-	0.67 *	−0.80 *	0.86 *	−0.75 *	−0.53 *
**EQ5D3L**	-	-	−0.48 *	0.69 *	−0.67 *	−0.30 *
**WPAI**	-	-	-	−0.68 *	0.51 *	0.37 *
**FACIT-F**	-	-	-	-	−0.84 *	−0.44 *

* *p* < 0.01, ^†^
*p* < 0.05.

**Table 3 jcm-08-01171-t003:** PROs in UC patients categorized by ClinPRO2 and mucosal healing.

	UC Patients Categorized by ClinPRO2 and Mucosal Healing			*p*-Value
	No Symptoms (n = 25)	Symptoms Present but MH (n = 15)	Symptoms Present and no MH (n = 50)	
Median age, years (IQR)	45 (32–63)	49 (35–64)	46 (31–54)	0.38
Male, n (%)	9 (36%)	7 (47%)	25 (50%)	0.24
Extensive UC, n (%)	14 (56%)	11 (73%)	30 (60%)	0.78
Current medication, n (%)				
5-ASA use	11 (44%)	8 (53%)	19 (38%)	0.56
Systemic corticosteroids	1 (4%)	2 (13%)	13 (26%)	0.06
Immunosuppressive drugs	7 (28%)	6 (40%)	12 (24%)	0.48
Biologic agents	10 (40%)	6 (40%)	24 (48%)	0.75
Patient-reported outcomes, median (IQR)				
SIBDQ ^†^	62 (58–64)	46 (33–56) *	40.5 (31.8–51) *	< 0.01
EQ5D3L ^†^	1 (0.827–1)	0.827 (0.761–1) *	0.816 (0.768–0.856) *	< 0.01
WPAI ^‡^	0 (0–0)	39.4 (16.9–71.7) *	62.5 (28.0–85.8) *	< 0.01
FACIT-F ^†^	47 (37–48.5)	24 (19–45) *	28.5 (18.8–40.3) *	< 0.01
HADS-depression ^‡^	2 (1–3)	6 (2–10) *	4 (2–7.3) *	< 0.01
HADS-anxiety ^‡^	4 (1–5)	8 (4–10) *	7 (4–10) *	< 0.01

* *p* < 0.05 when compared with no symptoms. SIBDQ: Short Inflammatory Bowel Disease Questionnaire, EQ5D3L: European Quality of Life 5-Dimensions 3-Level (time-trade-off), FACIT-F: Functional Assessment of Chronic Illness Therapy-Fatigue, WPAI: Work Productivity and Activity Impairment questionnaire (% of overall work productivity impairment), HADS: Hospital Anxiety and Depression Scale. ^†^ Higher values for SIBDQ, EQ5D3L, and FACIT-F scores indicate good quality of life. ^‡^ Higher values for WPAI, HADS-depression, and HADS-anxiety scores indicate poor quality of life.

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
