# Peer review of "The Combination of Patient-Reported Clinical Symptoms and an Endoscopic Score Correlates Well with Health-Related Quality of Life in Patients with Ulcerative Colitis"

_jcm, 2019, doi:10.3390/jcm8081171_

Round 1

Reviewer 1 Report

Thank you for submitting the manuscript to JCM. 

Sample size in this cross sectional study is barely adequate to reach a conclusion. Since the sample size is small the chances of error is high and the results could be due to coincidence alone. Especially in group of only 4 patients with no symptoms and no MH. But some of the findings showing very significant correlation which validate the findings, especially ClinPRO2 and ME correlation. 

Cross sectional study should represent the population but this study involved 96% Caucasians only. This makes it harder to extrapolate the findings to represent a larger population with different ethnicity. This is especially true when a study conducted in a single tertiary care center and also in a disease like ulcerative colitis

Really appreciate your efforts. Please accumulate further data on the same study and publish an update when you have ample size. Please try to incorporate other centers as well which will validate your findings better.

Author Response

Dear Editor,

Please find enclosed the edited manuscript in the attached file.

Manuscript ID:  jcm-556264

Title: “The Combination of Patient-Reported Clinical Symptoms and an Endoscopic Score Correlates Well with Health-related Quality of Life in Patients with Ulcerative Colitis”

Authors: Satimai Aniwan, David H Bruining, Sang Hyoung Park, Badr Al-Bawardy, Sunanda V. Kane, Nayantara Coelho Prabhu, John B. Kisiel, Laura E. Raffals, Konstantinos A. Papadakis, Darrell S. Pardi, William J. Tremaine, and Edward V. Loftus Jr.

We appreciate the comments/suggestions of the editors and reviewers.  Changes in the revised manuscript are highlighted in yellow.

Our point-by-point responses are below.

 Reviewer 1

1.     Sample size in this cross sectional study is barely adequate to reach a conclusion. Since the sample size is small the chances of error is high and the results could be due to coincidence alone. Especially in group of only 4 patients with no symptoms and no MH. But some of the findings showing very significant correlation which validate the findings, especially ClinPRO2 and ME correlation. 

Answer: We appreciate your comment. Because our primary objectives of this study were 1) to evaluate the correlation between the combination of ClinPRO2 and MES with HRQoL and 2) to evaluate the correlation between the SIBDQ and these HRQoL, we calculated sample size based on the assumption that the correlation coefficients (r-values) were ≥ 0.3.  Therefore, a sample size of at least 90 UC patients was required to detect these differences with a power of 80% at a two-sided significance level of 0.05.  We found that the correlations for all primary objectives were ≥0.3; therefore, the main conclusion is that in patients with ulcerative colitis, the combination of a ClinPRO2 and the MES had good correlation with the SIBDQ. In addition, SIBDQ was well correlated to the various HRQoL. We have already admitted this limitation on page 8 line 262 as follows: “Secondly, we initially attempted to assess the HRQoL of patients with no symptoms (MH and no MH). Unfortunately, only 4 patients with no symptoms did not have MH. This may limit the strength of the conclusion.”

2.     Cross sectional study should represent the population but this study involved 96% Caucasians only. This makes it harder to extrapolate the findings to represent a larger population with different ethnicity. This is especially true when a study conducted in a single tertiary care center and also in a disease like ulcerative colitis

Answer: Thank you for your concern. We have added this limitation into the Discussion (page 8 line 269).

“Fifth, the majority of our study population was Caucasian.  The generalizability of these data to different racial and ethnic populations may not be possible.”

3.     Really appreciate your efforts. Please accumulate further data on the same study and publish an update when you have sample size. Please try to incorporate other centers as well which will validate your findings better.

Answer: We are thankful for your suggestion. We have added this sentence into the Discussion (page 9 line 280).

“However, before the implementation of ClinPRO2 and HRQoL as a primary endpoint in clinical trials, the development of a reliable measurement of PROs from the patient’s perspective is necessary and requires future longitudinal studies from multinational regions.”

We would like to thank you once again for all of the kind suggestion and giving us an opportunity to resubmit this manuscript.

Yours Sincerely,

Edward V. Loftus, Jr., M.D.

Division of Gastroenterology and Hepatology

Mayo Clinic

200 First Street SW

Rochester MN 55905

USA

E-mail address: Loftus.Edward@mayo.edu

Reviewer 2 Report

This study aimed to evaluate the relationship between health-related quality of life (HRQoL) and the combination of a patient-reported clinical symptoms (ClinPRO2) and Mayo endoscopic subscore (MES) in patients with ulcerative colitis (UC).

Major points:

The limitations, described also by the authors, outweighs the strengths of the study. 

A validation cohort would be useful.

The clinical significance of the results is limited by the lack of an association with long-term clinical outcomes.

Minor points:

The lack of central reading for endoscopic findings is a limitation of the study.

A sensitivity analysis defining mucosal healing as MES 0 would be interesting.

Author Response

Dear Editor,

Please find enclosed the edited manuscript in the attached file.

Manuscript ID:  jcm-556264

Title: “The Combination of Patient-Reported Clinical Symptoms and an Endoscopic Score Correlates Well with Health-related Quality of Life in Patients with Ulcerative Colitis”

Authors: Satimai Aniwan, David H Bruining, Sang Hyoung Park, Badr Al-Bawardy, Sunanda V. Kane, Nayantara Coelho Prabhu, John B. Kisiel, Laura E. Raffals, Konstantinos A. Papadakis, Darrell S. Pardi, William J. Tremaine, and Edward V. Loftus Jr.

We appreciate the comments/suggestions of the editors and reviewers.  Changes in the revised manuscript are highlighted in yellow.

Our point-by-point responses are below.

Reviewer 2

This study aimed to evaluate the relationship between health-related quality of life (HRQoL) and the combination of a patient-reported clinical symptoms (ClinPRO2) and Mayo endoscopic subscore (MES) in patients with ulcerative colitis (UC).

Major points:

1.      The limitations, described also by the authors, outweighs the strengths of the study. A validation cohort would be useful. The clinical significance of the results is limited by the lack of an association with long-term clinical outcomes.

Answer: Thank you for your comment. We addressed these limitations into the Discussion (page 8 line 260).

“Further validated studies from other centers and other geographic regions would be useful. Additionally, further longitudinal studies are needed to explore whether improving disease activity improves the quality of life.”

Minor points:

2.      The lack of central reading for endoscopic findings is a limitation of the study.

Answer: We have added this sentence into the Limitation (page 9 line 271)

“Lastly, the lack of central reading for endoscopic findings could have introduced some bias; however, our two independent endoscopic readers were blinded to the clinical symptoms and HRQoL scores.”

3.      A sensitivity analysis defining mucosal healing as MES 0 would be interesting.

Answer: Thank you for your suggestion. However, there were only 10 patients with MES 0 out of 36 patients with MES 0-1; this would limit ability to draw conclusions from the statistical analysis.

We would like to thank you once again for all of the kind suggestion and giving us an opportunity to resubmit this manuscript.

Yours Sincerely,

Edward V. Loftus, Jr., M.D.

Division of Gastroenterology and Hepatology

Mayo Clinic

200 First Street SW

Rochester MN 55905

USA

E-mail address: Loftus.Edward@mayo.edu

Round 2

Reviewer 2 Report

The authors have adequately responded to the reviewers' comments.